# Effects of Strategic Supplementation with *Lupinus angustifolius* and *Avena sativa* Grains on Colostrum Quality and Passive Immunological Transfer to Newborn Lambs

**DOI:** 10.3390/ani12223159

**Published:** 2022-11-16

**Authors:** Giorgio Castellaro, Isaí Ochoa, Consuelo Borie, Víctor H. Parraguez

**Affiliations:** 1Department of Animal Production, Faculty of Agricultural Sciences, University of Chile, Santiago 8820808, Chile; 2Faculty of Veterinary Medicine and Zootechnics, National University Micaela Bastidas of Apurimac, Abancay 03000, Peru; 3Faculty of Veterinary Sciences, University of Chile, Santiago 8820808, Chile

**Keywords:** lupine grain, oat grain, immunoglobulins, merino ewes, late pregnancy

## Abstract

**Simple Summary:**

The quality and availability of grassland’s dry matter predispose ewes to undernutrition at the end of gestation, affecting the production and nutritional and immunological quality of colostrum. Furthermore, neonatal lambs are dependent on colostrum intake for immunity. Therefore, this study compared the effect of energy (oat grain) or protein (lupine grain) supplementation during late gestation on the chemical composition, energy value, and IgG content of the colostrum and the blood serum IgG concentration of newborn lambs. Sheep supplemented with oat grain had higher colostrum concentrations of protein and IgG and high IgG content in the blood serum of their lambs. Strategic oat grain supplementation during late gestation improved the nutritional and immunological quality of colostrum, positively affecting IgG transfer to lambs.

**Abstract:**

The aim of the present study was to evaluate the effect of two types of nutritional supplementation during late gestation on the chemical composition, energy value, and IgG concentration in the colostrum and the IgG concentration in the blood serum of lambs. Pregnant Merino Precoz ewes (*n* = 36) carrying single fetuses were used. Animals were kept grazing on the Mediterranean annual grassland. From day ~90 of pregnancy, animals were allocated into three groups: daily supplementation with oat grain or lupine grain and a control group without supplementation. Immediately after parturition, colostrum was collected from each ewe, and a blood sample was taken from the lambs 24 h after birth. For the evaluation of the chemical composition of the colostrum, an EKOMILK^®^ milk analyzer was used. The energy value of the colostrum was calorimetrically evaluated. IgG concentrations were measured by simple radial immunodiffusion. Data were analyzed by analysis of variance. Colostrum content of protein and non-fat solids was higher in the group supplemented with oat grain than in the lupine grain supplemented and control groups (*p* ≤ 0.05). In contrast, ewes supplemented with lupine grain had the highest concentration of fat in their colostrum (*p* ≤ 0.05). Oat grain supplementation resulted in higher concentrations of IgG, both in sheep colostrum and in the blood serum of their lambs (*p* ≤ 0.05), being higher than those observed in the lupine grain and control groups. Ewes that gave birth to male lambs had significantly higher concentrations of IgG in their colostrum compared to ewes that gave birth to females (*p* ≤ 0.05). The colostral IgG concentration positively correlated with the serum IgG concentration of the lambs (*r* = 0.32; *p* ≤ 0.05). The results indicate that the quality of colostrum and the immunological status of the newborn lambs can be improved by supplementation with oat grain.

## 1. Introduction

In ruminants, synepitheliochorial placentation prevents the transfer of maternal immunoglobulins to the fetus, and then the lambs are born with a null serum immunoglobulin G (IgG) concentration [1]. After birth, the lamb must absorb IgG from maternal colostrum before 18 h to acquire passive immunity [2]. Failure of passive transfer (FPT) is a condition predisposing the newborn to develop infectious diseases and death [3,4]. Low concentration of IgG in colostrum is considered one of the main factors contributing to the failure of passive transfer of immunity in lambs [4].

The feeding management of pregnant ewes during gestation should provide adequate energy and protein to support embryo–fetal growth, maintenance of the animal’s physiological needs, mammary gland growth, colostrum, and milk production [5]. Poor nutrition from mid to late pregnancy in ewes alters the quality and quantity of colostrum and decreases the birth weight of the newborn, which may have negative implications for lamb health and survival during the early postnatal period [6]. In contrast, feeding a diet that meets the energy requirement positively affects the fatty acid composition of colostrum, as well as IgG and insulin concentrations [7]. Therefore, nutritional strategies during the antepartum period influence the composition and immunological quality of colostrum [7].

In ewes, several nutritional factors have been associated with the concentration of fat, protein, and the yield of colostrum and milk. Among them, the most important are the energy balance, feed intake, size of ingested particles, neutral detergent fiber (NDF), non-fibrous carbohydrates (starch), and fatty acid composition of the feed [8].

Supplementation of ewes with crude protein at medium to high levels during the pre- and postpartum period increases the milk content of fat, protein, and total solids, as well as the lamb’s birth liveweight and daily liveweight gain [9]. On the other hand, it has been reported that excessive crude protein content in isoenergetic diets fed to ewes during late gestation increases birthweight and lambing difficulty but decreases colostrum yield and lambs’ survival [5]. In turn, undernutrition in late gestation reduces total IgG content in colostrum, compared to ewes receiving isoenergetic diets [6].

It has also been shown that ewe nutrition during late gestation influences udder development and mammary cell differentiation [10,11]. Ewes receiving protein and energy supplementation during the last stage of gestation produce a greater volume of colostrum but with a lower density (or lower viscosity) and higher lactose content [12].

In accordance with what was previously stated, we propose the following hypotheses: (I) the supplementation of pregnant ewes with grains containing high energy and protein during the last third of gestation improves the immunological and nutritional quality of the colostrum, compared to those not supplemented, and (II) the composition of the colostrum varies according to the energy or protein intake of the supplemented grains.

## 2. Materials and Methods

### 2.1. Study Area

The study was conducted on the Mediterranean annual range in the Small Ruminants and Dryland Grassland Section of German Greve Silva Experimental Station, Faculty of Agricultural Sciences, the University of Chile (33°28′ S; 70°51′ W; 470 m.a.s.l.) (Figure 1). The sector has a Mediterranean-type climate with an average annual temperature of 14.9 °C and an average annual rainfall of 285.6 ± 145.3 mm (years 1958–2021), concentrating 93.3% between April and September [13]. The soils belong to the Durixerolls family [14]. The predominant vegetation corresponds to a pseudo-savannah, with a shrub stratum dominated by *Vachellia caven* (Mol.) Seigler and Ebinger (“Espino”) and a herbaceous stratum composed of naturalized annual grasses (Genus: *Avena, Aira, Bromus, Hordeum, Vulpia,* and *Lolium*) [15,16].

A paddock of 27.7 ha of a naturalized grassland dominated by annual grasses and dicotyledonous herbs and with the presence in some sectors of *Phalaris aquatica* planted 35 years ago was used (Figure 2). During the trial period (May to July), the concentration of crude protein as a percentage of dry matter (DM) in the grassland varied between 18.3 and 21.4%, while the DM metabolizable energy concentration varied between 10.3 to 10.5 MJ/kg. The DM availability in the grassland, which was estimated using the comparative yield method [17], varied between 1324 to 2030 kg/ha.

### 2.2. Methods

A total of 50 healthy Merino Precoz ewes were used, and reproductive cycles were synchronized to facilitate the supplementation and limit the experimental period, avoiding possible differences in the quality of the pasture. For this purpose, progesterone intravaginal devices (CIDR G^®^, Pfizer, Santiago, Chile) were applied and maintained for 12 days. At the time of withdrawal, an i.m. dose of 150 IU equine Chorionic Gonadotrophin (eCG, Novormon^®^, Syntex, Buenos Aires, Argentine) was administered to each ewe. Two days later, ewes were exposed to 2 probed rams for three days. Pregnancy diagnosis was achieved by transrectal ultrasound exams one month after. Before starting the treatments, a second ultrasound exam was conducted to select 36 single-bearing ewes. At the beginning of the experiment, the animals were weighed (56.8 ± 4.8 kg; Pratley^®^ Precision Sheep Scale), and the body condition score (BCS) was assessed (2.91 ± 0.12; the scale of Russell et al.) [18]. Ewes were kept simultaneously grazing on the same natural grassland described previously during the entire pregnancy, and tap water was offered ad libitum. The trial began at approximately day 90 of gestation (range 89–91) by randomly assigning the ewes to the treatments: T1 (*n* = 12), supplemented with 230 g/day of lupine grain; T2 (*n* = 12), supplemented with 270 g/day of oat grain; a control group (C; *n* = 12) without supplementation. The chemical composition of the grains used as supplements and the pasture is shown in Table 1.

An initial period of 15 days for habituation to grain supplementation was implemented. The energy equivalent of the supplements was the same for both groups (2.75 MJ/ewe/day); it was determined as the average energy deficit that the ewes would theoretically have at the beginning of the study [21]. Every day at the end of the grazing period, the ewes were enclosed in individual pens, where they received the respective amount of supplement in accordance with their corresponding experimental group, consuming all the grains offered.

After parturition (0–4 h), colostrum samples were collected, previous administration of 5 IU oxytocin i.m., and teats were cleaned. Each ewe was hand milked, collecting a 20 mL colostrum sample for compositional and energetic analysis and a 5 mL sample for IgG determination. The colostrum samples were transported in refrigerated containers to the Animal Nutrition Laboratory (Faculty of Agricultural Sciences, University of Chile), where they were frozen at −20 °C until further analysis.

Twenty-four hours after birth, blood samples (5 mL) were collected from the lambs by jugular venipuncture [22], using syringes without anticoagulant. The samples were transferred to 1.5 mL Eppendorf tubes, leaving them to stand at room temperature for 6 h for coagulation. Subsequently, they were centrifuged using a microcentrifuge (SCILOGEX^®^ SCI-12 High Speed, SCILOGEX, LLC, Rocky Hill, CT 06067, USA) at 3500 RPM (616.3 RCF) for 15 min to separate the blood serum, which was stored in 1.5 mL cryovials at −20 °C until further analysis for IgG [23]. The lambs were kept with their mothers all the time; therefore, the consumed colostrum was taken by direct suckling on their mothers. Lambs were separated from their mothers only during the collection of colostrum and blood samples.

For the analysis of the chemical composition of colostrum, the samples were thawed and then placed in a container with water at 38 °C until they reached the analysis temperature (37–38 °C). The measurement was carried out using an EKOMILK M^®^ milk analyzer (Milk analyzer MIL-KANA KAM98-2ªA, Stara Zagora, Bulgaria), which determines the percentage of protein (P, %), fat (F, %), non-fat solids (NFS, %), and density (D, g/cm^3^). From this information, the percentage of total solids (TS, % = F + NFS) was calculated. An EKOMILK M^®^ milk analyzer was used reliably in both colostrum and milk analysis [24,25].

For the evaluation of the energy value, the colostrum samples were thawed at room temperature for 15 min, weighed, and placed in 100 mL beakers. The samples were dehydrated in a universal oven UF750 (Memmert Experts in Thermostatics, Schwabach, Germany) at 60 °C for 48 h. The colostrum energy value (EV, MJ/kg) was estimated using a calorimeter (Parr^®^ 6200 Isoperibol Calorimeter, Moline, IL, USA), placing a 0.5 g sample of dehydrated colostrum in the pump, allowing it to completely combust. Results were expressed both on an “as offered” basis (fresh colostrum) and on a “dry matter” basis. For this, it was necessary to determine the dry matter content of the colostrum, dehydrating the samples in an oven at 105 °C for 24 h.

To determine the IgG concentration in the colostrum and lambs’ blood serum samples, the simple radial gel immunodiffusion technique (SRI) was used based on the original protocol proposed by Fahey and McKelvey [26] and Mancini et al. [27], which was modified by Waldner and Rosengren [28]. A 1% agarose solution was prepared by adding 1 g of agarose to 100 mL of phosphate-buffered saline solution (pH 7.4) and boiling the mixture to dissolve the agarose. Then, the agarose solution was cooled by immersion in a water bath at 56 °C, and 3 g of rabbit anti-ovine IgG antibody (catalog N° 12-342, Sigma-Aldrich, Saint Louis, MO, USA) were added to it to achieve a concentration of 3%. Subsequently, the agar with the antibody was poured into 3 mL slides and left refrigerated in a humid chamber for 30 min. Finally, 8 wells, 2.5 mm in diameter, separated at 12 mm among them, were created with a micropipette tip.

Prior to the development of the SRI, both the colostrum and blood serum samples were thawed at room temperature for 15 min and diluted in phosphate-buffered saline (PBS, pH 7.4) at a ratio of 1:15 and 1:10, for colostrum and blood serum, respectively. Afterward, 4 µL of each diluted sample were deposited in each well of each SRI plate, and then it was placed in a humid chamber and incubated for 24 h at room temperature. After incubation, the diameter of the precipitation rings was measured. To determine the IgG concentrations of each sample, a calibration curve was established, using known concentrations of sheep IgG (1.25; 2.5; 5.0 and 10.0 mg/mL); catalog N° I5131, Sigma-Aldrich according to the protocol established by Waldner and Rosengren [28]. When necessary, more dilutions of the blood serum and colostrum samples were carried out to achieve the appropriate range of sensitivity of the radial immunodiffusion SRI plates.

Protocols for animal management, blood, and colostrum sampling were approved by the Animal Care and Research Advisory Committee at the Faculty of Agricultural Sciences, University of Chile (Certificate N° 20394-VET-UCH).

### 2.3. Statistical Analysis

For the statistical analysis of the nutritional composition, energy value and IgG concentration in colostrum, as well as for IgG concentration in the blood serum of the lambs, a completely randomized design with a factorial structure was used, which was evaluated by means of a general linear model of fixed effects with covariates [29]. The general expression was:𝑌_𝑖𝑗𝑘_ = *μ* + 𝑇𝑆_𝑖_ + SC_*j*_ + (TS·SC)_*ij*_ + *Covs* + 𝑒_𝑖𝑗𝑘_
where *Y_ijk_* represents the variable under analysis; *TS_i_* is the fixed effect of the type of supplementation (oat grain, lupine grain, and control without supplement); SC_*j*_ is the effect of the sex of the newborn lamb (male or female); (TS·SC)_*ij*_ is the interaction between the above factors. The effects of continuous or quantitative variables (*Covs*) were included as covariates, which in the case of colostrum traits were liveweight and BCS of the ewes immediately after parturition, while in the case of the analysis of IgG in lamb’s blood serum, the covariate was the birth weight. The variable *e_ijk_* is the experimental error. In the analysis of the nutritional composition and energy value of colostrum, the effect of the sex of the lambs and their interactions were not significant; therefore, they were removed from the statistical model. For the separation of means, Fisher’s LSD test was used. Significant differences were considered when *p* ≤ 0.05. Data are presented as means ± standard error of the mean.

In addition, a Pearson correlation matrix was calculated between all the variables analyzed in the study. Variables with a high and significant correlation were used to establish prediction equations through simple linear regression. To perform the above analyses, Statgraphics Centurion XVI^®^ software ver. 16.1.03 was used.

## 3. Results

All ewes gave birth to normal lambs, with an average body weight of 5.45 ± 0.13 kg, with no statistical differences between groups (*p* = 0.165).

### 3.1. Chemical Composition and Energy Value of Colostrum

The covariates, ewe’s liveweight and BCS at lambing, were not statistically significant for the variables of nutritional composition and energy value of colostrum (*p* > 0.05). The type of supplementation received by the ewes in the last third of gestation had a significant effect on the percentage of protein, fat, non-fat solids, and colostrum density. In the case of total solids, a tendency was obtained, although not significant (*p* = 0.085), to be higher in the group of sheep supplemented with oats (Table 2).

Ewes supplemented with oat grain had a higher percentage of P and NFS, followed by the control group and the group that received lupine grain. Regarding the F percentage in the colostrum, ewes supplemented with lupine grain had the highest concentrations (*p* = 0.005), while no differences were observed between the group that received oat grain and the control.

Colostrum density was affected by the type of supplementation (*p* ≤ 0.0001), where ewes receiving oat grain had the highest density, followed by the control group, and these two groups were higher than the group receiving lupine grain. The type of supplementation received by the ewes in the last third of gestation did not affect the energy value of the colostrum, both on a dry matter basis and as offered (*p* > 0.05).

### 3.2. Colostrum IgG Concentration

The covariate ewe’s liveweight and BCS at lambing, as well as the interaction between the fixed factors included in the model, were not significant (*p* = 0.220, *p* = 0.065, and *p* = 0.316, respectively). As shown in Table 3, the type of supplement received by the ewes in the last third of gestation significantly affected (*p* = 0.0003) the concentration of IgG in the colostrum. Supplementation with oat grain increased the concentration of IgG in the colostrum compared to the control group and the lupine grain group. There were no differences between these two latter treatments.

The sex of the newborn lamb had a significant effect on the concentration of IgG in the colostrum of the ewes (*p* = 0.0039), with the ewes that gestated male lambs having the highest concentrations of IgG (Table 4). There was no interaction (*p* > 0.05) between the categorical factors sex of the newborn lamb and the type of supplementation.

### 3.3. Concentration of IgG in the Blood Serum of Newborn Lambs

The birthweight, included as a covariate in the statistical analysis, did not have a significant effect on the serum IgG concentration of the lambs (*p* = 0.2787). The sex of the lambs also did not show a significant effect on the IgG serum concentration (females: 57.9 ± 3.2 mg/mL, males: 62.2 ± 3.4 mg/mL; *p* = 0.3744). There was no interaction between the fixed factors (sex of the newborn lamb and treatment; *p* = 0.7588).

The lambs from ewes supplemented with oat grain had a higher serum concentration of IgG (*p* = 0.0016); meanwhile, no differences in this trait were observed between the lupine grain and control groups (Table 5).

### 3.4. Correlations and Linear Regressions between Some Variables

Significant correlations were found between the chemical components and the density of the colostrum. Between F and P percentages, a moderate negative correlation was obtained. Similarly, the F percentage was negatively correlated with the percentage of NFS and colostrum density. On the other hand, colostrum protein was positively correlated with NFS content and colostrum density (Table 6).

Pearson’s correlation coefficient between maternal colostrum and newborn lambs’ serum IgG concentrations was positive and moderate (*r* = 0.3235; *p* ≤ 0.05). Due to the high and positive correlation coefficient obtained between P and IgG in colostrum, it was possible to determine a linear equation that predicts the magnitude of the change in IgG concentration in colostrum (IgG, mg/mL) as a function of its protein concentration (P, %) (Figure 3).

The calculated equation was significant, as well as its coefficients (*p* ≤ 0.001). The coefficient of determination adjusted for freedom degrees (R^2^) was 65.5%, with an estimated standard error (SEE) of 13.7 mg/mL. The above equation makes it possible to predict the immunological quality of colostrum from its protein percentage with an acceptable degree of certainty.

A positive, high, and significant relationship between colostrum density and its IgG concentration was also observed (Table 6). Thus, it made it possible to determine a linear equation that predicts the magnitude of the change in colostrum IgG concentration (IgG, mg/mL) as a function of its density (D, g/cm^3^) (Figure 4).

In this case, the determined regression equation was also highly significant, as were its coefficients (*p* ≤ 0.0001). The adjusted coefficient of determination value was 63.4%, and the estimated standard error (SEE) was 14.07 mg/mL. The previous equation also allows the prediction of the immunological quality of the colostrum from its density with an acceptable degree of certainty.

## 4. Discussion

### 4.1. Nutritional Composition and Energy Value of Colostrum

Differences in the chemical composition of colostrum according to the breed of sheep have been reported. In this regard, Ciuryk et al. [30], working with Polish Merino ewes, indicate a composition of 11.1% of fat, 21.2% of protein, and 32.2% of dry matter, values that are higher than those found in the present study for the Merino Precoz bred. In contrast, colostrum from Corriedale ewes presents lower values than those of the present study [31]. However, reported values described for other sheep breeds coincide with ours, among them the Dorset [32], Suffolk × Greyface, Texel × Greyface [33], and Rahmani [34].

Supplementation with oat grain in the prepartum diet was positive for improving the percentage of protein, non-fat solids, and colostrum density, being higher compared to the control group and ewes supplemented with lupine grain. Diets high in nonstructural carbohydrates provided a sufficient amount of energy for the ruminal microbiota without the need to break down proteins as an energy source. This resulted in greater availability of metabolizable proteins for the animal and, therefore, greater availability of proteins for milk synthesis or other tissues. Likewise, it might be expected that with lower production of ammonia, the toxicity threshold will not be exceeded [35,36]. Grains such as corn, barley, and oats ferment in the rumen, but an important fraction also provides starch for digestion in the small intestine, where it is absorbed as glucose [12]. It has been reported that the energy concentration of the diet is positively associated (*r* = 0.64) with the concentration of protein in milk [37]. This results from increased propionate production in the rumen, leading to a decreased rate of utilization of amino acids destined for gluconeogenesis, which are used in the synthesis of milk proteins [38]. In this regard, Banchero et al. [35] indicate that when cereal grains with high energy content in the form of starch are used as supplements during late gestation, an increase in colostrum production (90–185% above the control non-supplemented group) is observed. The variation observed between the different types of cereal grains used could be attributed to differences in the amount of starch that escapes ruminal fermentation [39].

Ewes fed diets with a high percentage of degradable protein, as occurred with lupine grain supplementation in the present study, could increase the concentration of ruminal ammonia and thus induce an increase in ammonia in the blood, reaching toxic levels in some cases [35], especially if there is not an additional source of non-fibrous carbohydrate in the diet [12]. Although lupine grain also provides high amounts of metabolizable energy, given its high protein degradability (as shown in Table 1), higher concentrations of ruminal ammonia could be expected, making it necessary to invest greater energy expenditure in liver detoxification. This would explain, at least partially, the lower colostrum concentrations of proteins and NFS observed in the ewes supplemented with lupine grain compared with those supplemented with oat grain or with the control.

On the other hand, it has been reported that ewes fed diets low in non-fibrous carbohydrates (starches) mobilize body fat by releasing non-esterified fatty acids (NEFA) into the bloodstream, which are used as precursors of synthesis of milk fat, thereby increasing its percentage in colostrum [40]. This could explain the high fat colostrum concentration present in the group supplemented with lupine grain compared to the other two groups.

The type of supplementation received by the ewes did not affect the energy value of the colostrum. This could be explained by the fact that the energy value would be determined by the percentages of fat and non-fat solids and their corresponding caloric values. Although the group of ewes supplemented with oat grain produced colostrum with a higher concentration of protein, it also had a lower concentration of fat. The opposite occurred in the group supplemented with lupine grain, which partially compensated for the energy differences. In this regard, Abd-Allah [34] reported that the energy value of colostrum does not vary when pregnant ewes are subjected to different nutritional levels during late gestation.

### 4.2. Concentration of IgG in Ewe’s Colostrum

The concentration of IgG in the colostrum of Merino Precoz ewes has not been previously reported. The average colostrum IgG concentrations obtained in the present study were 100.7 ± 2.9 mg/mL. These concentrations are similar to those reported in other breeds such as Columbia (115.1 ± 10.1 mg/mL) [41], Dorper (78.3 ± 8.3 mg/mL) [42], Australian Merino (112.1 ± 23.9 mg/mL) [42], and Crossbreed ewes in Brazil (142.5 mg/mL) [33], but not those of superior Shaul (62.9 ± 2.5 mg/mL) [4], Lori Bakhtiyari ewes (48.8 ± 2.1 mg/mL) [4,43], and Merino (35.7 ± 2.5 mg/mL) [2].

Nutritional status during the last two-thirds of pregnancy affects lamb birth weight, mammary gland development, and colostrum components [6]. In contrast, several studies have shown that the ewe´s BCS does not affect the concentration of IgG in the colostrum [42,43,44], a situation that was also observed in our trial. However, differences in diet composition can modify the IgG content of colostrum. For instance, feeding with high-energy supplements that meet nutritional requirements during the last period of gestation increases colostrum IgG [7], in addition to improving the colostrum yield [45], which partly coincides with our results. In our study, it was determined that ewes supplemented with a source of a high content of non-fibrous carbohydrates (*Avena sativa* grain) results in higher concentrations of IgG in the colostrum compared to ewes that did not receive this kind of supplementation. Supplementation with high non-fibrous carbohydrate grains decreases progesterone and growth hormone levels while increasing insulin and insulin-like growth factor type I (IGF-I) levels, resulting in an increased ability of the udder to produce colostrum [46,47]. Consistent with the above, and although insulin was not evaluated in the present study, this type of supplementation could have increased insulin, which in turn could have decreased gluconeogenesis and thus made more amino acids available for protein synthesis. These factors could be related to the higher concentrations of IgG obtained in the colostrum of sheep supplemented with oat grain.

In cattle, immunoglobulins present in colostrum are derived from those circulating in the maternal blood and are actively transported to the mammary gland through binding to neonatal FcRn receptors [48,49]. In addition, the increased expression of FcRn receptors in the mammary gland has been shown to correlate with the period of increased IgG transfer to ewe´s colostrum [49]. The FcRn receptor has been cloned in sheep, and its distribution in the mammary gland in the peripartum suggests that it fulfills the same role described for FcRn in cows [50]. Fallah et al. [51], in their experiment of supplementing ewes (Lori-Bakhtiari breed) with lycopene and corn grain, mention that an increase in circulating IgG concentrations has, consequently, an increase in IgG in colostrum at 6 h postpartum, which may support what was obtained in our experiment when supplementing with cereal grains.

Higher circulating levels of IgG in the blood and greater mammary development, together with increased expression of FcRn receptors, could explain the differences between the treatments evaluated in the present trial. We did not evaluate these variables, but they are certainly of interest for future studies.

### 4.3. Concentration of IgG in Lamb’s Blood Serum

Concentrations of IgG in the lamb’s blood serum at 24 h after birth ranged between 21.51 and 81.25 mg/mL. Hunter et al. [41] reported that these concentrations could be in a range of 0 to 102 mg/mL at 24 h after birth. This high variability in the levels of IgG in the lamb’s blood serum could be explained by the concentration of IgG in the colostrum, as well as by the amount of colostrum consumed by the newborn lamb. Thus, the colostrum IgG concentration, as a result of the amount and source of energy consumed by the ewe, has tremendous relevance for the passive immunity acquired by the lambs. In this sense, Campion et al. [52] reported that the increase in net energy in the pregnant ewe led to a linear increase in IgG concentration in the lamb’s blood at 24 h postpartum. In addition, Fallah et al. [51] reported a substantial increase in blood IgG in Lori- Bakhtiari lambs, reaching a concentration of 25 mg/mL 24 h after birth, as a result of maternal supplementation with lycopene and corn grain. However, these IgG concentrations are lower than those obtained in our experiment.

Regarding the association between IgG concentrations in ewes’ colostrum and lamb´s blood serum, Vatankhanh [43] reported a correlation of medium magnitude between both variables (*r* = 0.43, *p* ≤ 0.05), but slightly higher than that obtained in our study. A similar situation occurs in other ruminant species. For instance, Jaster [53] showed that calves feeding on colostrum with a high IgG concentration (84 mg/mL) led to high IgG values in their blood (38.7–45.7 mg/mL). In contrast, when calves consumed colostrum with low IgG concentration (31.2 mg/mL), the IgG concentration in blood serum was low (10–13.8 mg/mL). A similar situation was observed in our study, where those lambs that consumed colostrum with a higher concentration of IgG presented higher concentrations of IgG in the blood serum (*r* = 0.324; *p* ≤ 0.05). However, this kind of correlation has not been found in other sheep studies [4,42]. The IgG concentration achieved in the blood serum of lambs is not only affected by the IgG concentration in maternal colostrum but also by the amount of colostrum consumed in the first hours of life [54], as well as by the intestinal absorption of colostrum [55]. In this sense, a probable weakness of our study is that we could not measure the colostrum consumed by the lambs, so this factor cannot be ruled out as influencing the serum concentration of IgG in the lambs.

On the other hand, a breed factor has been pointed out as having a great influence on the variability of blood concentrations of IgG. In this regard, Ahmad et al. [22] reported a mean of 28.9 mg/mL 24 h after birth in lambs of the Pak-Karakul breed, while in Lori-Bakhtiari lambs, Vatankhah et al. [43] reported levels between 5.97 to 8.32 mg/mL, at 36 h after birth. In Shaul lambs, Nikbakht et al. [56] reported levels of 31.2 ± 14.1 and 34.2 ± 12.4 mg/mL in males and females, respectively, in measurements conducted 3 days after birth, although without significant differences (*p* = 0.53). However, this statement seems premature since there is not enough information to compare the different sheep breeds at the same postpartum times.

## 5. Conclusions

The chemical composition of colostrum can be modified by the type of diet offered to the ewes during late gestation. In this sense, strategic nutritional supplementation may be beneficial to obtain a colostrum of better nutritional and immunological quality. Supplementation with grains with a high energy value, such as oat grain, improves the colostrum’s nutritional composition and its IgG content. This is not only important to ensure better nutrition for the lambs but also to strengthen the passive immunity of the lambs. Moreover, colostrum IgG concentration can be predicted from its protein concentration and/or its density with an acceptable degree of precision, so these variables could be used as indicators of the immunological quality of the sheep colostrum. Unexpectedly, it was observed that ewes that gestated male lambs presented higher IgG colostrum concentrations compared to those that gestated females, a situation that deserves to be investigated.

## Figures and Tables

**Figure 1 animals-12-03159-f001:**
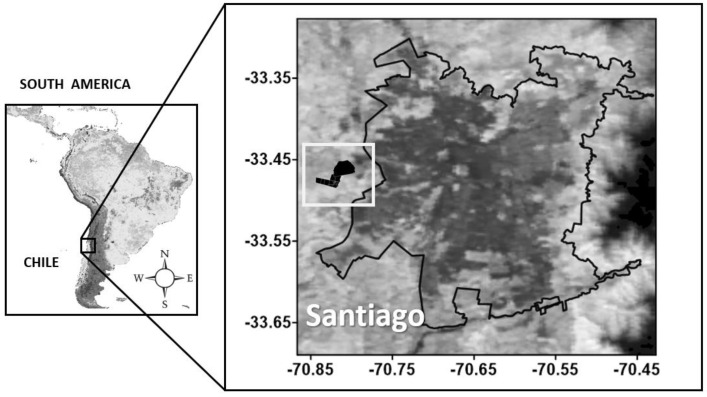
Location of the area where the study was carried out.

**Figure 2 animals-12-03159-f002:**
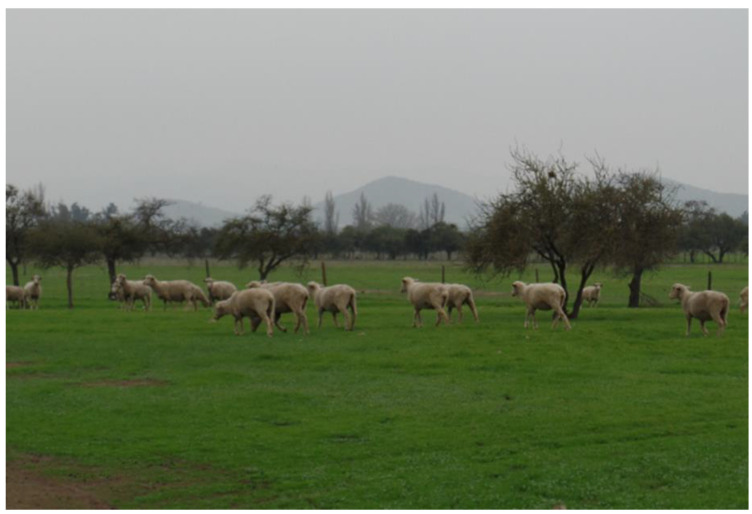
Merino precoz ewes grazing on Mediterranean annual grassland where the experimental study was carried out (fall-winter season).

**Figure 3 animals-12-03159-f003:**
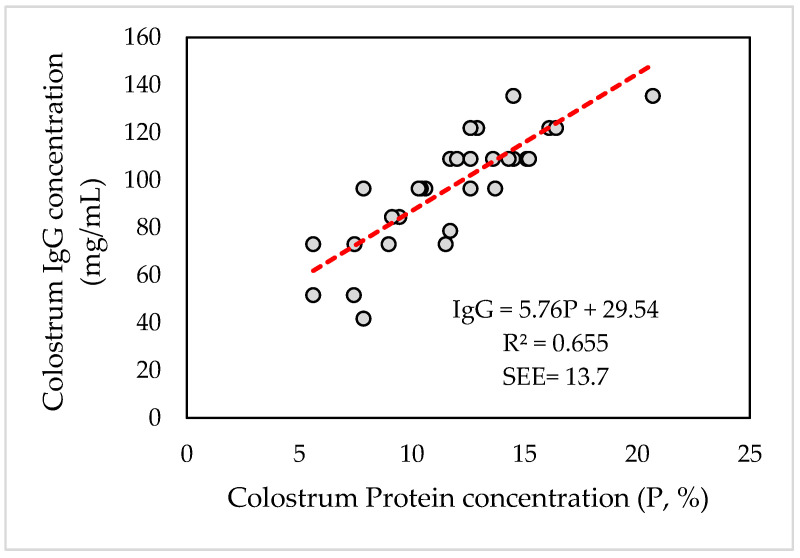
Relationship between IgG concentration (mg/mL) and protein concentration (P, %) in the colostrum of Merino Precoz ewes.

**Figure 4 animals-12-03159-f004:**
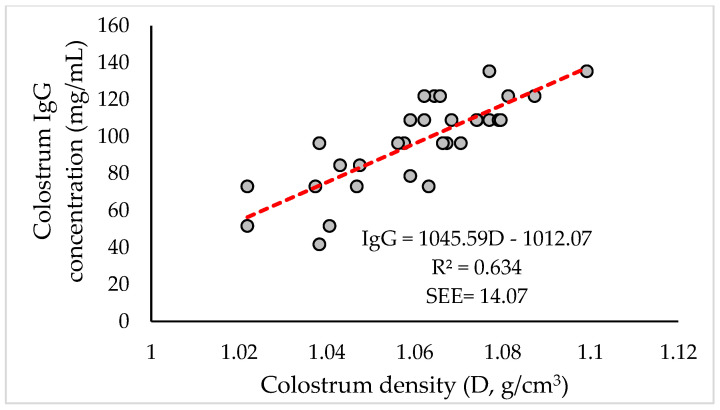
Relationship between IgG concentration (mg/mL) and density (D, %) in the colostrum of Merino Precoz ewes.

**Table 1 animals-12-03159-t001:** Chemical composition of the grains used as a supplement and the pasture used as a base diet (MS: dry matter; ME: metabolizable energy; CP: crude protein; EfD: effective degradability of crude protein; NDF: neutral detergent fiber; ADF: acid detergent fiber).

Treatment	MS ^1^ (%)	ME ^2^ (MJ/kg)	CP ^1^ (%)	EfD ^2^ (%)	NDF ^1^ (%)	ADF ^1^ (%)
Lupine grain	86.80	13.72	29.45	80.10	25.10	18.70
Oat grain	89.81	11.55	11.52	71.70	33.38	26.40
Pasture	30.00	10.40	19.90	75.60	40.40	29.52

^1^ Chemical analysis performed according to the techniques proposed by A.O.A.C. [19]. ^2^ Estimated using equations proposed by SCA [20].

**Table 2 animals-12-03159-t002:** Percentage of protein (P), fat (F), non-fat solids (NFS), total solids (TS), and density (D), determined in the colostrum of Merino Precoz ewes (least squares mean ± standard error), supplemented with different grains.

Components of Colostrum	Supplementation Type		
	Lupine Grain	Oat Grain	Control
P (%)	9.05 ± 0.68 ^c^	14.93 ± 0.68 ^a^	11.28 ± 0.68 ^b^
F (%)	12.57 ± 0.97 ^a^	8.23 ± 0.97 ^b^	8.41 ± 0.97 ^b^
NFS (%)	15.16 ± 0.82 ^c^	21.68 ± 0.82 ^a^	17.57 ± 0.82 ^b^
TS (%)	27.73 ± 1.20	29.91 ± 1.20	25.98 ± 1.71
D (g/cm^3^)	1.046 ± 0.004 ^c^	1.078 ± 0.004 ^a^	1.060 ± 0.004 ^b^
EV (MJ/kg, DM basis)	30.23 ± 0.92	28.69 ±0.92	28.15 ± 0.92
EV (MJ/kg, offered basis)	8.53 ± 0.67	8.30 ± 0.67	8.77 ± 0.67

Means with different superscript letters in the same row indicate significant differences among treatments (LSD Fisher, *p* ≤ 0.05).

**Table 3 animals-12-03159-t003:** Colostrum IgG concentration (means least squares ± standard error) in Merino Precoz ewes (least squares mean ± standard error) under different supplementation types.

Ewe’s Supplementation Type	IgG (mg/mL)	Range (mg/mL)
Lupine grain	86.0 ± 5.1 ^b^	41.7–121.9
Oat grain	118.8 ± 5.2 ^a^	96.4–135.4
Control	97.4 ± 5.1 ^b^	73.0–121.9

Means with different superscript letters in the same column indicate significant differences among treatments (LSD Fisher, *p* ≤ 0.05).

**Table 4 animals-12-03159-t004:** Concentration of IgG in ewe’s colostrum, according to the sex of the newborn lamb (means least squares ± standard error).

Newborn Lamb’s Sex	IgG (mg/mL)	Range (mg/mL)
Male	110.1 ± 4.3 ^a^	73.0–135.4
Female	91.4 ± 4.1 ^b^	41.7–135.4

Means with different superscript letters in the same column indicate significant differences between treatments (LSD Fisher, *p* ≤ 0.05).

**Table 5 animals-12-03159-t005:** Concentration of IgG in the blood serum of newborn lambs at 24 h postpartum (least square means ± standard error), under different maternal supplementation modalities, during the last third of gestation.

Ewe’s Supplementation Type	IgG (mg/mL) in Blood Serum	Range (mg/mL)
Lupine grain	51.2 ± 3.8 ^b^	21.5–64.2
Oat grain	72.0 ± 3.8 ^a^	64.3–81.3
Control	57.7 ± 3.8 ^b^	27.8–72.6

Means with different superscript letters in the same column indicate significant differences among treatments (LSD Fisher, *p* ≤ 0.05).

**Table 6 animals-12-03159-t006:** Pearson’s correlation matrix for the components of ewe’s colostrum.

	P (%)	F (%)	NFS (%)	TS (%)	D (g/cm^3^)	IgG (mg/mL)
P (%)	1	−0.365 *	0.990 **	0.857 **	0.974 **	0.816 **
F (%)		1	−0.380 *	0.162 ^ns^	−0.530 **	−0.308 ^ns^
NFS (%)			1	0.849 **	0.986 **	0.816 **
TS (%)				1	0.749 **	0.698 **
D (g/cm^3^)					1	0.803 **
IgG (mg/mL)						1

(*) *p* ≤ 0.05; (**) *p* ≤ 0.01; ns: non-significant.

## Data Availability

The data on which these results are based can be requested directly from the authors.

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
