# Peer review of "Effects of Strategic Supplementation with Lupinus angustifolius and Avena sativa Grains on Colostrum Quality and Passive Immunological Transfer to Newborn Lambs"

_animals, 2022, doi:10.3390/ani12223159_

Round 1

Reviewer 1 Report

 Description

 Suggestion

Article title

Ok

Simple Summary and Abstract

Ok

Introduction

Placenta of ruminants: Synepitheliochorial

L-45 - epitheliochorial

Placenta of Ruminants: Synepitheliochorial (Wooding, 1992).

Wooding, F.B.P. Current Topic: The Synepitheliochorial Placenta of Ruminants: Binucleate Cell Fusions and Hormone Production. Placenta, v.13 p.101-113.

Materials and Methods

L-91 …average annual rainfall of 285.6±145.3 mm (years 1958-2021)

It is true? Only 285.6 mm per year?

L – 111 – ….body condition score (BCS)...

What is the method adopted to measure the body condition of the ewes?

L-125 - …Each ewe was hand milked, collectiong a 20 mL of colostrum sample ….

Please provide more detail on how the colostrum samples were taken? Was the milking complete? How many kg of colostrum was obtained from each ewe? Is sampling performed after full milking? Detail how the colostrum homogenization was carried out to obtain the samples. Was oxytocin used to get all the colostrum?

L – 130 – Twenty-four hours after birth, blood samples (5mL) were colleted ….

Describe in detail the management given to newborns, how did they feed on colostrum? Did each newborn lamb ingest its mother's colostrum or was it from a colostrum bank? How much colostrum did each lamb ingest from the ewe's first postpartum milking Was colostrum given as a function of birth weight or were they breastfed ad libitum?

L – 139 - … using a EKOMILK M® ….

How reliable is the use of the EKOMILK device for milk analysis? Reinforce the citation of an author who has used this equipment and validated its use in scientific research.

Results

ok

Discussion

L – 387 to 391 - Please adjust the discussion by referring to the species studied.

… in lines 387 to 391, this discussion is based on two references (references 43 and 44) addressing receptor (FcRn) in the bovine mammary gland and not in ewes, as it should be. Suggestion: cite a source that shows that the receptor (FcRn) in ruminants are the same or cite sources referring to the ewes mammary gland.

L – 411 … 24 hours after birth (reaching 25 mg(mL), as a result….

… 24 hours after birth (reaching 25 mg/mL), as a result….

Conclusions

L-433- 445 - Conclusions

Please adjust the conclusion based on the two statements of the hypotheses. Supplementation with oat grains increases the concentration of IgG in colostrum. And does lupine grain supplementation also increase the concentration of IgG in colostrum?

Reviewer 2 Report

This kind of research is essential to improve animal production because of the importance for newborn ruminants to acquire enough immunoglobulins through colostrum. However, there is a lack of several essential evaluations that should have been done on the feed, ewes, and lambs. That decreases the reliability of the interpretation of the results in the discussion.

Reviewer 3 Report

This manuscript describes high starch versus high protein supplementation of pregnant ewes grazing annual meadow on colostrum quality and passive transfer. Overall, the experiment is scientifically sound and well described in the manuscript. The English grammar needs improvement - there are improper use of prepositions, etc.

Specific comments:

L3 - delete 'it effect on'

L25 - elsewhere in the manuscript it says that treaments started at 90 days of gestation

L62 - this is a 1 sentence paragraph

L113 - were all ewes grazing the same pasture at the same time. I assume they were, but this should be explicitly clear since individual ewe is the experimental unit

L116 - please include in text or table the nutritional composition of the supplements

L124 - this phrase is unclear. What does 'previous teats' mean?

L142 - including both protein and non-fat solids in the equation means that you are double counting protein because protein is included in non-fat solids

L159-165 - I don't see where you added the unknown samples in the SRI procedure. please improve the clarity of this description

L187 - typically P-values are written as decimals rather than percents

L194 - if sex effect is important for colostrum IgG, why not include sex in the model for colostrum nutritional components?

L206 - The lambing date or back calculated actual day of gestation at start of treatments should be included to ensure there were no differences among treatments

L209 - since covariates are not shown in tables, please include actual p-values rather than just P>0.05

Table 1 - change to 'grains' in title

Table 1 - the values for protein and fat are rather high compared to other studies. This is discussed later

Table 1 - do not show superscript if no statistical difference

L231 - although not less than 0.05, it appears BCS likely influenced colostrum IgG, which is similar to other studies. This is not discussed in the Discussion section

Table 4 - one thing that could have improved the study was measuring the quantity of colostrum produced (milk use and bottle feed lambs) so that lamb serum IgG could be adjusted for quantity consumed. You do not know whether differences in IgG in blood is due to colostrum IgG concentration, colostrum intake, or ability of lamb to absorbed IgG (late gestation heat stress in dairy cows has been shown to decrease absorptive capability of calf small intestine). This should be discussed in the paper.

Table 5 - why was lamb serum IgG not included in this table

L290 - please do not use the term 'highly significant' because there is no such thing

L322-328 - this section is discussing the effect of highly fermentable starch on rumen parameters and implications for increased colostrum nutrient concentrations. However, it is very poorly explained. This paragraph needs to be rewritten.

L342 - yes, the high protein intake and degradability would result in high ammonia levels and more energy for conversion to urea, but I don't think you can say that 'much of it' would be used. 

L346-350 - in this paragraph you are telling me that lupine is high in starch, and the previous paragraph you told me that lupine was low in starch

L364-365 - show the IgG values from these studies so the reader can compare with your values

L375-379 - I dont understand this sentence or the logic behind it. Please further or better explain what you mean. Also, us the treatment names rather than terms like 'last group'

L423 - in line 416 you told me this correlation was 0.43. Which is correct?

L425 - I would describe this as a breed factor not a racial factor

L434-425 -  this implies that both lupine and oat increased composition of colostrum, but only oat did. please re-write these sentences.

Reviewer 4 Report

Review Animals-1919421

Title: Effects of strategic supplementation with Lupinus anguistifolius and Avena sativa grains on colostrum quality and it effect on passive immunological transfer to newborn lambs.

General considerations: The question to be answered by the article has lost relevance in the last 15 years. Studies with supplementation of pregnant females addressed questions about fetal programming.

The hypotheses described in the article present a certain obviousness about the results that will be achieved.

The lack of information on the chemical composition of the treatments (lupine grain and oat) compromised the quality of the article.

Due to this last factor, the opinion is for the rejection of the article.

Simple Sumary: There is a good description, making it possible to understand the question and the result.

Abstract: L30 – Delete the phrase  “Data were analyzed by ANOVA.” This detail is unnecessary.

Keywords: Delete “supplementation”. The word is already in the title.

1.       Introduction: The articles cited in the introduction could be more recent. Of the 14 articles, 2 are more than 6 years old; 1, 10 years and 11 more than 15 years.

2.       Materials and Methods

- L102-105 – “During the trial period (May to July), the concentration  of crude protein as a percentage of dry matter (DM) in the grassland varied between 18.3 and 21.4%, while the DM metabolizable energy concentration varied between 10.3 to 10.5 104 MJ/kg. The DM availability in the grassland varied between 1324 to 2030 kg/ha.” - Describe in detail how the results were obtained.

- L110-111 – “Pregnant single-bearing merino precoz ewes (n=36; 56.8 ± 4.8 kg of liveweight; 2.91 ±  0.12 of body condition score (BCS)”. Describe how and where the ewes were weighed and which BCS scale was used.

- L98 and L107 - I believe that Figures 1 and 2 are unnecessary.

-L114 - Were the sheep all inseminated on the same day?

- L116 - What is the chemical composition of feeds?

- L176 - What experimental design was used?

3.       Results

4.       Discussion

The effect of supplementation on the Nutritional composition and energy value of colostrum and Concentration of IgG in ewe’s colostrum needs further discussion. For example, for the effect of sheep breeds on the variables, it is necessary to inform the difference between the breeds.

Another factor that compromised the discussion was the absence of measurements of serum biochemistry in sheep (eg ammonia and glucose) that could support some of the authors' assumptions.

The lack of information on the chemical composition of the treatments (lupine grain and oat) compromised the quality of the article.

Due to this last factor, the opinion is for the rejection of the article.

Round 2

Reviewer 2 Report

The manuscript has been improved. However, there are still some details that must be written more clearly. Also, some statements made by the authors need to be discussed differently according to what was actually measured in this study.

Author Response

Dear reviewer....

The responses to your comments are attached as a pdf file

Reviewer 4 Report

I am satisfied with the explanations that have been given. The article is accepted for publication.

Author Response

Dear Reviewer 4:

We appreciate your valuable and detailed review that has allowed us to substantially improve the draft of our research work

Sincerely